

# Effect of chronic unpredicted mild stress-induced depression on clopidogrel pharmacokinetics in rats

Xueyao Jiang[1,*], Jing Wu[1,*], Boyu Tan[2], Sulan Yan[3], Nan Deng[4] and Hongyan Wei[4]

[1] Hunan Normal University, Changsha, Hunan, China
[2] Department of Pharmacy, Shanghai Children's Hospital, Shanghai Jiao Tong University, Shanghai, China
[3] Department of Cardiovascular, The First Affiliated Hospital of Hunan Normal University, Hunan Provincial People's Hospital, Changsha, Hunan, China
[4] Department of Pharmacy, The First Affiliated Hospital of Hunan Normal University, Hunan Provincial People's Hospital, Changsha, Hunan, China
[*] These authors contributed equally to this work.

Corresponding author
Hongyan Wei,
weihongyan@hunnu.edu.cn,
weihy2001@126.com

## ABSTRACT

**Background.** Clopidogrel is widely used to prevent and treat cardiovascular atherosclerosis and thrombosis. However, disturbance in the expression and activity of liver cytochrome metabolic enzymes significantly changes clopidogrel efficacy. Therefore, the effect of chronic unpredictable mild stress (CUMS)-induced depression on the expression of liver cytochrome metabolic enzymes and clopidogrel pharmacokinetics in rats were explored.

**Methods.** Nine different CUMSs were selected to establish a rat model of depression. Open field experiment and sucrose preference test were applied to explore the depressive behaviors. The concentration of serotonin in the cortex of depressed rats was determined using enzyme linked immunosorbent assay (ELISA). All rats were given 10 mg/kg clopidogrel orally after 12 weeks, and blood samples were collected at different time points. The clopidogrel concentration and CYP2C19/ CYP2C9 activity in rat liver microsomes were assayed by high performance liquid chromatography-tandem mass spectrometry (HPLC-MS/MS). The rat liver drug enzymes expression was determined by Real-Time Quantitative Reverse Transcription PCR (RT-qPCR).

**Results.** Open field experiment and sucrose preference test indicated the successful construction of the CUMS-induced depression model. The concentration of serotonin in the cortex of depressed rats decreased by 42.56% (**$p < 0.01$). The area under the curve of clopidogrel pharmacokinetics decreased by 33.13% (*$p < 0.05$) in the depression rats, while distribution volume and clearance increased significantly (**$p < 0.01$). The half-time and distribution volume did not significantly differ. The CYP2C19 and CYP2C9 activity of liver microsomes in the CUMS-induced depression group were significantly higher than that in the control group (**$p < 0.01$). *CYP2C11* and *CYP1A2* mRNA expression up-regulated approximately 1.3 - fold in the depressed rat livers compared with that in the control, whereas that of *CYP2C13* was down-regulated by 27.43% (**$p < 0.01$). *CYP3A1* and *CYP2C12* expression were slightly up-regulated, and that of *CES1* did not change.

**Conclusions.** These results indicated that CUMS-induced depression altered clopidogrel pharmacokinetics, and the change in *CYP450* activity and expression in depressed rat livers might contribute to the disturbance of clopidogrel pharmacokinetics.

## INTRODUCTION

Clopidogrel is a P2Y12 antiplatelet inhibitor that is indispensable in treating and preventing atherosclerosis and thrombosis (*Patti et al., 2020*) and can reduce the recurrence rate of atherosclerotic thrombotic events in patients with acute coronary syndrome (ACS). Patients with ACS undergoing percutaneous coronary intervention (PCI) need to take clopidogrel for 6–12 months to prevent in-stent thrombosis (*Braun & Kassop, 2020*). However, ischemic or bleeding events are widespread in patients after PCI, which may be partly due to insufficient or excessive platelet inhibition at the standard clopidogrel treatment dosage (*You et al., 2020*). Genetic polymorphism affects the absorption, distribution and metabolism of clopidogrel. Furthermore, epigenetic modification and P2Y12 receptor also interfere with its antiplatelet activity (*Zhang et al., 2017*). In addition, recent evidence suggests that non-genetic factors such as disease complications of depression and drug interactions could alter the antiplatelet effect of clopidogrel (*Serbin, Guzauskas & Veenstra, 2016*; *Black & Held, 2017*).

Several clinical studies have shown that the rate of psychological stress response in cardiovascular disease patients is high during the lifetime, especially in female (*Bucciarelli et al., 2020*). Stress actives the hypothalamus pituitary adrenal, which is easy to produce insomnia or high blood pressure, and even coronary spasm and hypertensive crisis in severe cases (*Martinac et al., 2014*). In addition, psychological stress produces emotional disorders, mainly manifested as depression. Depressed patients reduce medication compliance, leading to high recurrence rate and mortality of acute coronary syndrome. The depression can also cause metabolic disorder, increase blood lipid level, shorten platelet coagulation time and damage vascular endothelial cells. These factors aggravate the coronary atherosclerosis, leading to plaque rupture and bleeding (*Black & Held, 2017*).

It is well known that clopidogrel is safe in healthy volunteers, but it can induce recurrence of gastric ulcers in high-risk patients (*Chan et al., 2005*). Clinical studies have shown that clopidogrel increases the risk of bleeding in patients with comorbid depression of acute myocardial infarction (*Labos et al., 2011*). The long-term psychological stress hormone greatly increase the probability of comorbid depression in ACS patients (*Aladio et al., 2021*). When patients with comorbid depression need to take antidepressants, drug interaction increases the adverse drug reaction of clopidogrel (*Delavenne et al., 2013*; *Yuet et al., 2019*).

As an inactive prodrug, clopidogrel becomes an active product through liver CYP450 metabolism (*Jiang et al., 2015*). Previous studies have found that depression caused by chronic unpredictable stress could perturb in the expression and activity of CYP450 in the liver, leading to change the process of drug metabolism *in vivo* (*Xia et al., 2016*). Therefore, the clopidogrel pharmacokinetics in patients with depression and whether the activity and expression of liver drug enzymes are disrupted need to be further studied. Thus, this study aimed to establish a chronic unpredictable stress (CUMS)-induced depression rat model

to explore the metabolism of clopidogrel in depressed rats. This would guide adjusting the clinical dosage of clopidogrel and provide theoretical basis and experimental reference for the individualized application of clopidogrel in patients with depression.

## MATERIALS & METHODS

### Chemicals and reagents

Clopidogrel standard sample (No: 20101201, 98.00%) was purchased from molnova Co., Ltd. (Shanghai, China), and ticlopidine (internal standard, IS; No: SLBD1982, ≥99%) was purchased from Sigma Trading Co., Ltd. Clopidogrel hydrogen sulphate tablets were obtained from Sanofi Pharmaceuticals Co., Ltd (Hangzhou, China). Substrates of S-mephenytoin, 4-hydroxymephenytoin, tolbutamide, and 4-hydroxytolbutamide substrates (≥95%) were purchased from Sinco Pharmachem (Middletown, DE, USA). Diazepam (internal standard, IS; 171225-201304) was purchased from the China Institute of Food and Drug (China). High-performance liquid chromatography (HPLC)-grade methanol and ethyl acetate were purchased from Merck (Kenilworth, NJ, USA). 5-Hydroxytryptamine (5-HT) ELISA Kit of rat was purchased from CUSABIO (Houston, TX, USA). TRIzol Reagent was purchased from Sigma-Aldrich Trading Co., Ltd. (Shanghai, China). The PrimeScript TM RT Master Mix (No:RR047A) and SYBR Premix Ex Taq$^{TM}$ (No: RR420A) were purchased from TaKaRa Bio Inc (Dalian, China).

### Animals

Female SPF SD rats (5 weeks old, approximately 200 ± 20 g weight; Animal Quality Certificate: 2019-0004) were purchased from Hunan SLAC Laboratory Animal Co., Ltd. (Changsha, China). The rats were kept at the Laboratory Animal Center, Hunan Provincial People's Hospital, Changsha, China (Animal Experiment License: SCXK 2020 - 260), in a well-ventilated and clean laboratory environment (temperature 22−25 °C, relative humidity 50%–70%). All rats had free access to obtain food and water for 1 week. Then rats were randomly divided into two groups (each group comprising nine rats): control and depression groups.

### CUMS-induced depression model

The depression group was provided feed and water *ad libitum* in a cage alone for 12 weeks. The stressors included restraint (activity restriction in a bottle, 1 h), hot water swimming (45 °C, 5 min), cold water swimming (4 °C, 5 min), tail clipping (one cm apart from thetail, 1 min), cage stilting (45 °C, 24 h), horizontal shaking (10 min), damp padding (24 h), noise interference (10 min), and day/night inversion (24 h). Each stressor was randomly used 8–9 times (*Wei et al., 2017*), and the same stressor was not applied consecutively to avoid rat's prediction. After the stress treatment, the rats were moved back. The control group rats were provided feed and water *ad libitum* for 12 weeks without any stressor. The experimental grouping and depression modeling are shown in flow chart (Fig. 1).

### The sucrose preference test

The sucrose preference test was performed to explore depression-like behavior based on the rat's natural preference for sweets taste (*Duca, Swartz & Covasa, 2014*). Before testing,

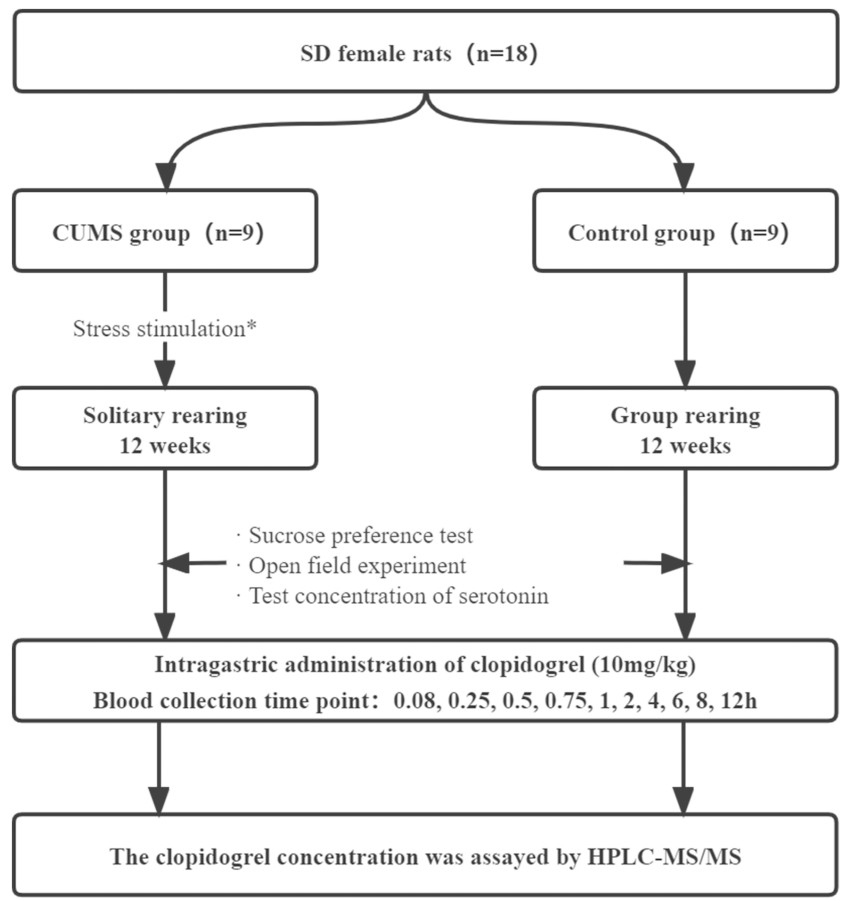

**Figure 1 Schematic diagram of experimental design of depression disturbing clopidogrel metabolism in female rats.** *: The stressors included restraint (activity restriction in bottle, 1 h), hot water swimming (45 °C, 5 min), cold water swimming (4 °C, 5 min), clip tail (1 cm apart from the tail,1 min), cage stilting (45 °C, 24 h), horizontal shaking (10 min), damp padding (24 h), noise interference (10 min), and day/night inversion (24 h).

all rats were kept in a single cage. Two bottles of 1% sucrose solution were provided to each cage of rats for adaptive training for 24 h. Thereafter, one bottle of 1% sucrose solution was replaced with ordinary drinking water, and the adaptive training was conducted for a further 24 h. Following this acclimation, the rats were deprived of water and feed for 24 h, and the formal test was started at 8:00 pm on the fourth day. Each cage of rats was provided with two bottles of pre-weighed solution (one bottle of 1% sucrose solution and one bottle of ordinary drinking water), without feed deprivation. After 12 h, they were weighed again to calculate the consumption from each bottle of solution using the following equation:

sugar water preference rate(%) = [1% sucrose water consumption/

(1% sucrose solution consumption + ordinary drinking water consumption)] × 100.

### Open-field test (*Katz, Roth & Carroll, 1981*)

Before and after the model establishment, the open-field behavior of each rat was analyzed using a SuperMaze Video Analysis System (Shanghai XinRun Information Technology Co., Ltd., Shanghai, China). The apparatus comprised an opaque box (100 cm × 100 cm × 50 cm). The open-field area was divided into $33 \times 33$ cm$^2$ equal-sized squares. The average moving speed, total moving distance, freezing time, and central area entry times were determined as indices of locomotion activity and exploratory behavior. The test was conducted in a quiet room in the evening (5:00 pm–00:00 am) for 10 min.

### Determination of serotonin (5-HT) level in rat brains

SD rats were euthanized by cervical dislocation after completing the pharmacokinetic experiment. Brain samples were collected from each rat and frozen in liquid nitrogen as described previously method (*Wu et al., 2020b*). When 5-HT was detected, the cortex of rats were first transferred to a refrigerator at −20 °C for freezing, and then thawed about at 20 °C. Subsequently, the left cerebral cortex (100 mg) was excised, cut into small pieces, and put into a tissue grinder, after which 1 mL 1× phosphate-buffered saline (PBS) was added into the grinder to make a homogenate, which was then stored −20 °C overnight. After repeated freezing and thawing twice to destroy the cell membrane, the tissue homogenate was centrifuged at 5,000 × g at 2−8 °C for 5 min, and the supernatant was used for 5-HT determination. The 5 - HT levels were measured using ELISA (CUSABIO, US) according to the manufacturer's instructions.

### Dosage regimen and sample collection

All rats were fasted overnight but with free access to water, then administered 10mg/kg of clopidogrel orally (*Wu et al., 2020a*). Blood samples were collected via the eye canthusat at 0.07, 0.25, 0.5, 0.75, 1, 2, 4, 6, 8, and 12 h after isoflurane inhalation anesthesia. Unfortunately, three rats died of bleeding during the experiment. The blood samples were centrifuged at 3,000 × g for 5 min to obtain plasma, which was stored at −80 °C for further analysis.

### Determination of clopidogrel concentration

The clopidogrel concentration in rats' plasma were was determined using high-performance liquid chromatography-tandem mass spectrometry (HPLC-MS/MS) method (*Xiao et al., 2019*). A total of 50 μL plasma sample and 10 μL ticlopidine (5 ng/mL) were added to 0.2 mL ethyl acetate in a centrifuge tube, vortex-mixed for 3 min and centrifuged at 14,000 × g for 10 min. The supernatant was evaporated to dry. Next, the residue was redissolved in 100 μL methanol by vortex-mixing for 1 min and centrifuged at 16,000 × g for 1 min. Thereafter, 2.0 μL of the supernatant was injected into the HPLC-MS/MS system for analysis. The HPLC instrument equipped with verse-phasecolumn was connected to a triple quadrupole tandem MS with an electrospray interface. The system comprised an LC system (Nexera UHPLC/HPLC; Shimadzu, Kyoto, Japan) with a Triple Quad 6500 MS system (AB Sciex, Framingham, MA, USA). Chromatographic separation was achieved using an ACE Excel 5 SuperC18 column (SN: A191749; 150 × 2.0 mm; Advanced Chromatography Technologies Ltd, Aberdeen, UK). The mobile phase comprised methanol (A) and water

(0.1% ammonium formate, B) with a flow rate of 0.3 mL/min. The gradient elution included 30%A (0–0.5 min), 5%A (0.5–1.5 min), 5%A (1.5–2.0 min), 30%A (2.0–2.5 min), and 30%A (2.5–4.0 min). The MS conditions were optimized as follows: a positive electrospray ionization (ESI$^+$) mode, capillary at 5,500 V, an ion-spray gas temperature of 500 °C, a gas flow rate of 11 L/min, and nebulizer at 10 psi. The parameters of clopidogrel and Sit were as follows: fragmentary voltages of 70 V and 66 V, collision energy of 20 units and 23 units, respectively. The multiple-reaction monitoring mode was selected to quantify clopidogrel and Sit, for which the precursor-to-product ion transitions were 322.1 → 212.0 and 264.3 → 154.1, respectively. The MassHunter Workstation software (Version B.06.00; Agilent Technologies, Santa Clara, CA, USA) was used to collect and process data.

## Determination of CYP2C19 and CYP2C9 activity in liver microsomes

The control and CUMS-induced depression rats were fasted for 12 h and killed by cervical dislocation before the liver was excised, rinsed with ice-cold normal saline (0.9% NaCl, w/v), weighed to 1 g, and homogenized with 4.0 mL in a 0.05 M Tris-HCl buffer (pH 7.4). The homogenate was centrifuged at 10,000 × g at 4 °C for 20 min, and the supernatant was further centrifuged at 105,000 × g at 4 °C for 60 min. Afterward, the helvus sediment was reconstituted with 1.0 mL Tris-HCl buffer and stored at −80 °C for further analysis. The total protein concentration of liver microsomal enzymes was determined using a Bicinchoninic Acid Protein Assay Kit. In addition, the CYP450 activity of the liver microsomes was determined using a cocktail of probe drugs in which the activities of CYP enzymes were assessed by determining probe drug production. Tolbutamide (TOL) or mephenytoin (MPT) were metabolized to 4-OH-TOL or 4-OH-MPT by CYP2C9 or CYP2C19, respectively, and these products were determined by HPLC-MS (*Lasker et al., 1998*).

The liver microsome working solution was thawed at 4 °C, then absorbed 100 μL, probe substrate 100 μL, phosphate buffer 250 μL in 5 mL brown centrifuge tube, vortex oscillation for 1 min, incubation in 37 °C water bath for 5 min, adding NADPH 50 μL to start the reaction, and incubation in 37 °C water bath for 20 min. After the reaction, the incubation system was placed in an ice bath, then 1.4 mL of ethyl glacial acetate and 20 μL internal standard (diazepam; 5 μg.mL$^{-1}$) were added, subjected to vortex oscillation for 2 min, and centrifuged at 3,500 × g at 4 °C for 10 min. Thereafter, the supernatant was sucked into a 1.5 mL brown centrifuge tube and placed in a vacuum drying oven (80 °C, 0.9–1.0 MPa) for evaporation. Next, the supernatant was redissolved with 200 μL methanol, vortex oscillated for 1 min, centrifuged at 14,000 × g for 2 min, and 5 μL of the supernatant were sucked into an injection bottle for analysis. The whole experiment was conducted in darkness.

Chromatographic separation was achieved using an ACE Excel 5 SuperC18 column (2.0 mm × 150 mm, 1.8 μm, SN: A191749; Advanced Chromatography Technologies Ltd.). The mobile phase contained 0.1% formic acid water (A) - methanol (B) with a flow rate of 0.3 mL/min. The gradient elution included 50% A (0–2.4 min), 5% A (2.4–3.2 min), and 50% A (3.2–4.5 min). The optimized MS conditions were as follows: ESI$^+$ mode, a source injection voltage of 5,500 V, collision gas of 10 psi, inlet voltage of 10 V, temperature

of 500 °C, curtain gas of 30 psi, and multi-reaction detection scanning (MRM). The parameters of 4-OH-TOL and 4-OH-MPT included a fragmentary voltage of 100 V and collision energies of 24 and 23 units, respectively. The monitoring ions were set as m/z 287.3 → 170.8 for 4-OH-TOL and 235.2 → 150.2 for 4-OH-MPT.

### mRNA expression of Cytochrome P450 by RT-qPCR

Rat livers were collected after the pharmacokinetic experiment, washed with 0.9% NaCl solution, and stored at −80 °C until further use. Total RNA in rats' livers was extracted using TRIzol Reagent (Sigma Trading Co., Ltd.). The RNA concentration was determined using a NanoDrop Spectrophotometer (Thermo Fisher Scientific, Waltham, MA, USA) at an absorbance of 260 nm. Total RNA was transcribed to cDNA using a PrimeScript RT reagent Kit with gDNA Eraser (Takara Bio Inc.), and RT-qPCR was performed using an ABI 7500 Real-time PCR system with SYBR Green Quantitative PCR Master Mix. To normalize the mRNA expression, the housekeeping gene $\beta$-actin was used as an external standard. The primer sequences used were shown in Table 1. The relative amount of each mRNA was calculated using the $2^{-\Delta\Delta Ct}$ method (*Wei et al., 2017*).

### Statistical analysis

Statistical and pharmacokinetics parameter analyses were performed using SPSS 22.0 (SPSS Inc., Chicago, IL, USA) and DAS 3.0 software (Shanghai University of Traditional Chinese Medicine, Shanghai, China). Differences between the two groups were analyzed using an independent samples $t$-Test. Data are expressed as the mean ± standard deviation (SD). *$p < 0.05$ or $< 0.01$ was considered statistically significant.

## RESULTS

### Validation of depression model

The sucrose preference test, behavior scores of open-field test, and 5-HT levels were shown in Fig. 2. The sucrose preference and behavior scores were the same between the two groups before the depression model establishment, but changed significantly after the depression model establishment (**$p < 0.01$, Fig. 2). The sucrose preference, average moving speed, total moving distance, and central area entry times decreased from 81.71 ± 10.73 to 68.16 ± 9.33%, from 104.21 ± 38.69 to 7.32 ± 4.05 mm/s, from 3,137.46 ± 1,166.84 to 438.95 ± 243.13 cm, and from 6.89 ± 4.49 to 0.22 ± 0.67, (**$p < 0.01$) in the depression group, respectively. However, the freezing time increased from 271.39 ± 50.74 to 523.53 ± 31.50s in the depression group. There were no change for these in the control group ($p > 0.05$). After the model establishment, the cortex 5-HT levels of rats were also reduced significantly (4.52 ± 0.99 vs. 2.59 ± 0.31 ng/mL) compared with the two groups.

### Clopidogrel assay and pharmacokinetic parameters

Clopidogrel and IS retention times were 2.74 min and 1.21 min, respectively (Fig. 3). No significant interference or ion suppression peaks were observed in the retention regions of both compounds. The correlation coefficient of calibration curves showed

**Table 1  Primer sequences for quantitative real-time polymerase chain reaction.**

| Gene symbol | GeneBank accession No | Prime sequence for qRT-PCR | Amplicon size (bp) |
|---|---|---|---|
| CYP2C11 | NM_019184.2 | F:5′-GGACATCGGCCAATCAATAAA-3′<br>R:5′-TGCCCATCCCAAAAGTCC-3′ | 269 |
| CYP2C13 | NM_138514.1 | F:5′-CACTTTCTTCAGTTCCTCCCACTT-3′<br>R:5′-CCTCATCTTCATTTCTGTTTTCTGG-3′ | 81 |
| CYP3A1 | NM_012105.2 | F:5′-GGAAATTCGATGTGGGAGTGC-3′<br>R:5′-AGGTTTGCCTTTCTCTTGCC-3′ | 79 |
| CYP1A2 | NM_012541.3 | F:5′-CAGTGGAAAGACCCCTTTGTGT-3′<br>R:5′-CAAGCCGAAGAGCATCACC-3′ | 102 |
| CYP2C12 | NM_031572.2 | F:5′-TTCTCAGCAGGAAAACGGAAATG-3′<br>R:5′-TCGATGTCCTTTGGATCAGACAG-3′ | 122 |
| CES1 | NM_001190375.1 | F:5′-CTACCCACCTATGTGCTCCC-3′<br>R:5′-GCCCAGGCGATACTGAATGAC-3′ | 274 |
| $\beta$-actin | NM_031144.2 | F:5′-CCTGACCGAGCGTGGCTA-3′<br>R:5′-CCACAGGATTCCATACCCAGGAA-3′ | 242 |

linearity ($R^2 > 0.99$) by weight of $1/x^2$ over the concentration range of 0.35–49.93 ng/mL. The lowest limit of quantitation (LLOQ) was 0.35 ng/mL, and the matrix effects were approximately 96%. Intra-day accuracy and precision were 99.79%-106.46% and 2.18%−6.34%, respectively; for inter-day, these were 97.46%-106.25% and 2.54%−8.10%, respectively (Table 2). The concentration–time parameters were calculated using the non-compartmental analysis method (Fig. 4 and Table 3). The area under curve ($AUC_{0-\infty}$), peak time ($T_{max}$), peak concentration ($C_{max}$), and mean residence time ($MRT_{0-t}$) in the depression group decreased by 33.13% (from $82.56 \pm 22.22$ to $55.21 \pm 21.91$), 10.52% (from $0.38 \pm 0.24$ to $0.34 \pm 0.17$), 17.71% (from $12.42 \pm 7.17$ to $10.22 \pm 4.63$) and 11.89% (from $11.27 \pm 3.03$ to $9.93 \pm 2.09$), respectively, but only the area under curve ($AUC_{0-\infty}$) had a notable significance ($^*p < 0.05$). The distribution volume (Vz/F; $1841.58 \rightarrow 3718.20$ L kg$^{-1}$) and clearance (CLz/F; $128.31 \rightarrow 219.87$ L h kg$^{-1}$) increased significantly ($^*p < 0.01$). No statistically significant difference was observed in half-time ($T_{1/2}$) between the two groups.

## Determination of CYP450 activity and relative expression

The retention times of 4-OH-TOL and 4-OH-MPT were 3.38 min and 1.81 min, respectively. The CYP2C19 and CYP2C9 activities of liver microsomes in the depression group were significantly increased than those in the control group (Fig. 5). The activity of CYP2C19 in depression group increased approximately 3-fold from $0.025 \pm 0.004$ to $0.072 \pm 0.005$ ng/(min mg protein) ($^{***}p < 0.001$). The CYP2C9 activity also varied significantly, increasing from $0.948 \pm 0.112$ to $1.528 \pm 0.151$ ng/(min mg protein) ($^{**}p < 0.01$).

The expression of *CYP450* in rat livers was analyzed using RT-qPCR. The relative expression of *CYP2C11* mRNA increased by 22.60% in CUMS group, while that of *CYP2C13*

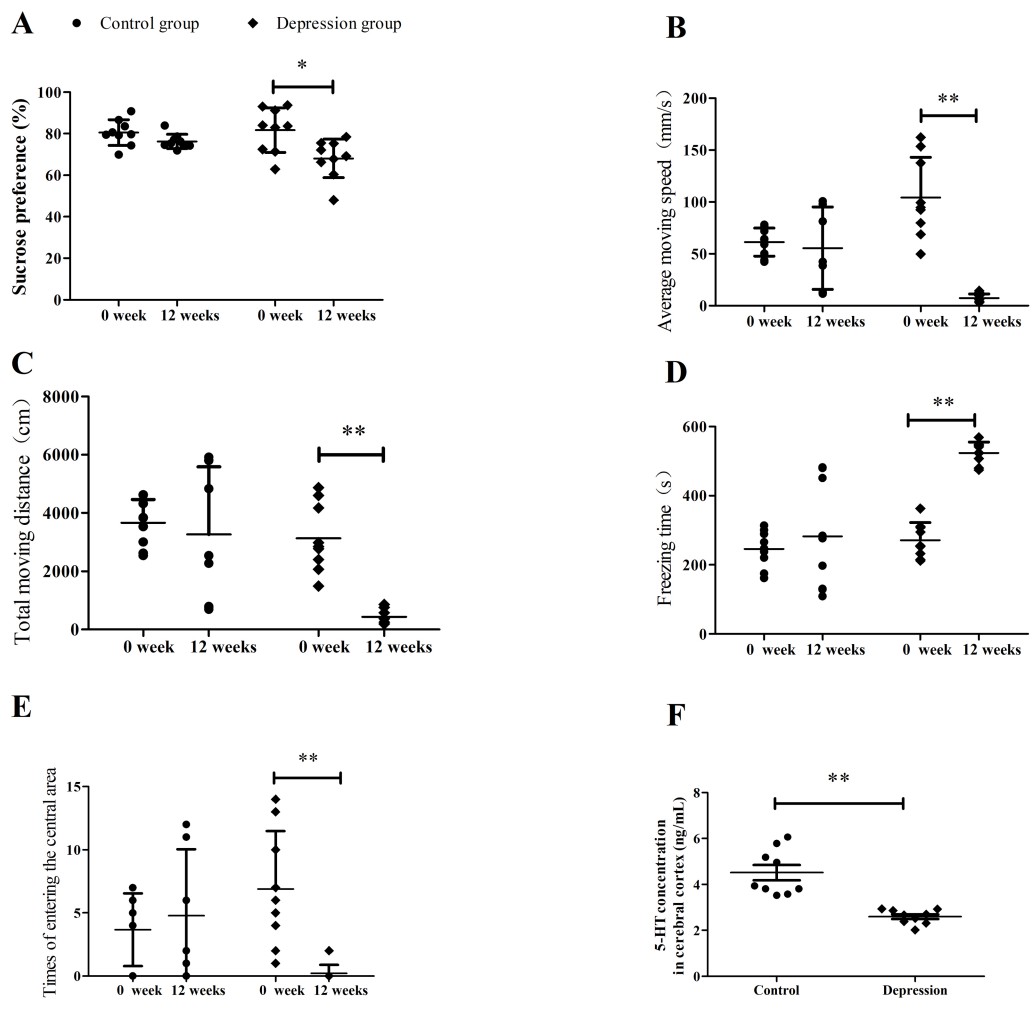

**Figure 2** **Results of sucrose preference test, depressed behaviors and serotonin level in rats.** (A) Sucrose preference test, (B) average moving speed, (C) total moving distance, (D) freezing time, (E) times of entering the central area, and (F) cortex 5-HT level. ($n = 9$, mean $\pm$ SD). $*p < 0.05, **p < 0.01$: CUMS compared with the control group.

mRNA decreased by 27.43%. *CYP1A2* increased by 34.39% ($**p = 0.01$) in CUMS group compared with control group. *CYP3A1* and *CYP2C12* mRNA relative expression increased slightly between the CUMS group, but did not reach the significance. *CES1* mRNA relative expression did not change in CUMS group.

## DISCUSSION

CUMS has been widely used, reliable, and effective rodent model of depression (*Antoniuk et al., 2019*). An ideal animal model of depression must be able to simulate the behavior of patients with depression and psychological stress factors that cause depression. In modern society, people suffer from all kinds of stress, such as the reduction of living space, working overtime, traffic and living noise, drastic changes in the environment and climate, and

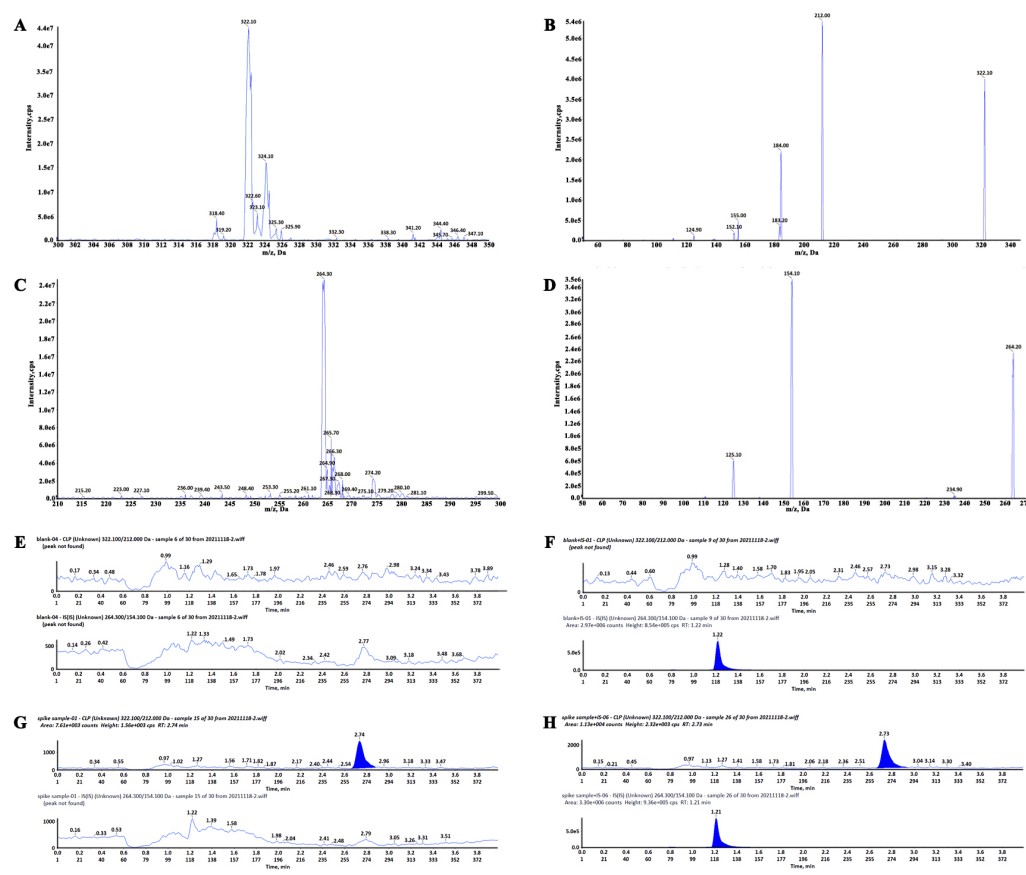

**Figure 3   Full-scan and product ion scan mass spectrogram and chromatograms of clopidogrel and IS.** (A) Full-scan mass spectrogram of clopidogrel, (B) product ion scan mass spectrogram of clopidogrel, (C) full-scan mass spectrogram of IS, (D) product ion scan mass spectrogram of IS, (E) chromatograms of blank plasma, (F) blank plasma spiked with clopidogrel, (G) chromatograms of IS, (H) plasma sample with IS and clopidogrel.

**Table 2   Evaluation on analytical methods of clopidogrel ($n = 6$, mean ± SD).**

| Theoretical concentration (ng/mL) | Intra-day | | | Inter-day | | |
|---|---|---|---|---|---|---|
| | Measured concentration (ng/mL) | Precision (RSD %) | Accuracy (%) | Measured concentration (ng/mL) | precision (RSD %) | Accuracy (%) |
| 0.35 | 0.35 ± 0.02 | 6.34 | 100.48 | 0.34 ± 0.03 | 8.10 | 97.46 |
| 0.69 | 0.70 ± 0.03 | 3.94 | 100.72 | 0.69 ± 0.03 | 4.51 | 100.4 |
| 5.55 | 5.91 ± 0.13 | 2.18 | 106.46 | 5.90 ± 0.15 | 2.54 | 106.25 |
| 49.93 | 49.82 ± 1.63 | 3.28 | 99.79 | 49.71 ± 2.02 | 4.07 | 99.56 |

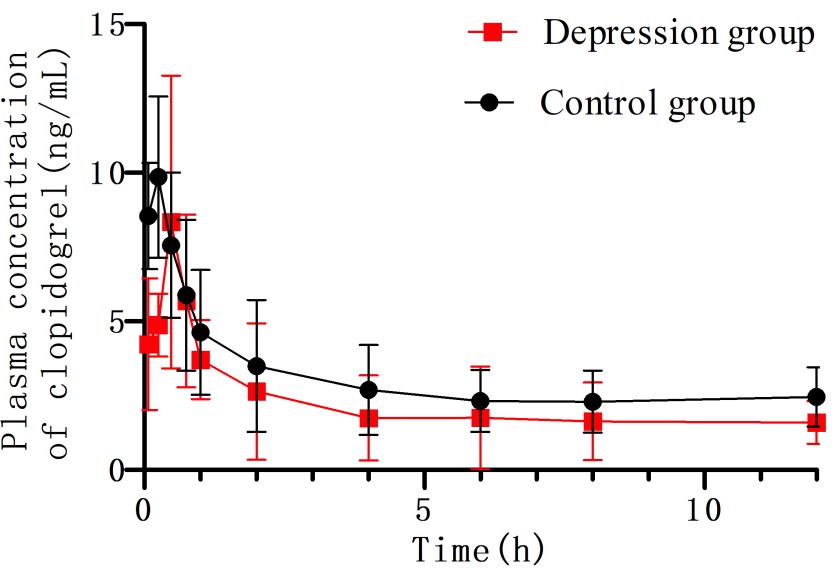

**Figure 4 The concentration–time curve of clopidogrel in plasma after oral 10 mg/kg ($n = 6$, mean ± SD).** The area under curve in the depression group significantly decreased compared with the control group ($82.56 \rightarrow 55.21 \, \mu g \, h \, L^{-1}$). The distribution volume (Vz/F) and clearance (CLz/F) increased ($1,841.58 \rightarrow 3,718.20 \, L \, kg^{-1}$), ($128.31 \rightarrow 219.87 \, L \, h \, kg^{-1}$), respectively.

**Table 3 Pharmacokinetic parameters of clopidogrel in rats' plasma.**

| Parameters | Control group | Depression group |
| --- | --- | --- |
| $t_{1/2}$ (h) | $10.17 \pm 3.14$ | $11.45 \pm 3.67$ |
| $T_{max}$ (h) | $0.38 \pm 0.24$ | $0.33 \pm 0.17$ |
| Vz/F ($L \, kg^{-1}$) | $1,841.58 \pm 561.08$ | $3718.20 \pm 1770.18$[**] |
| CLz/F ($L \, h \, kg^{-1}$) | $128.31 \pm 32.39$ | $219.87 \pm 76.63$[**] |
| $C_{max}$ ($\mu g \, L^{-1}$) | $12.42 \pm 7.17$ | $10.22 \pm 4.63$ |
| $AUC_{(0-\infty)}$ ($\mu g \, h \, L^{-1}$) | $82.56 \pm 22.23$ | $55.21 \pm 21.91$[*] |
| $MRT_{(0-t)}$ (h) | $11.27 \pm 3.03$ | $9.93 \pm 2.09$ |

**Notes.**

[*], $p < 0.05$; [**], $p < 0.01$: depression group compared with the control group ($n = 6$, mean ± SD).

so on. CUMS model is based on the effects of various stresses faced by humans. These mild stresses have a reasonable theoretical basis, and the modeling method is related to the causes of clinical depression. The forced swimming test is outstanding in all aspects of depressive behavior, but this stress method is too simple and fixed in intensity, which may make the animals tolerant and predictable in the process of modeling. After that, *Katz, Roth & Carroll (1981)* used chronic unpredictable severe stress methods. For example, strong noise, strong light stimulation, long-term restraint, *etc.* Compared with previous studies, these stresses can make experimental animals unable to predict the occurrence of stress, but the more intense stress model does not meet the reason for people's depression (*Katz, Roth & Carroll, 1981*). *Papp, Willner & Muscat (1991a)* adjusted Katz's CUMS model. This

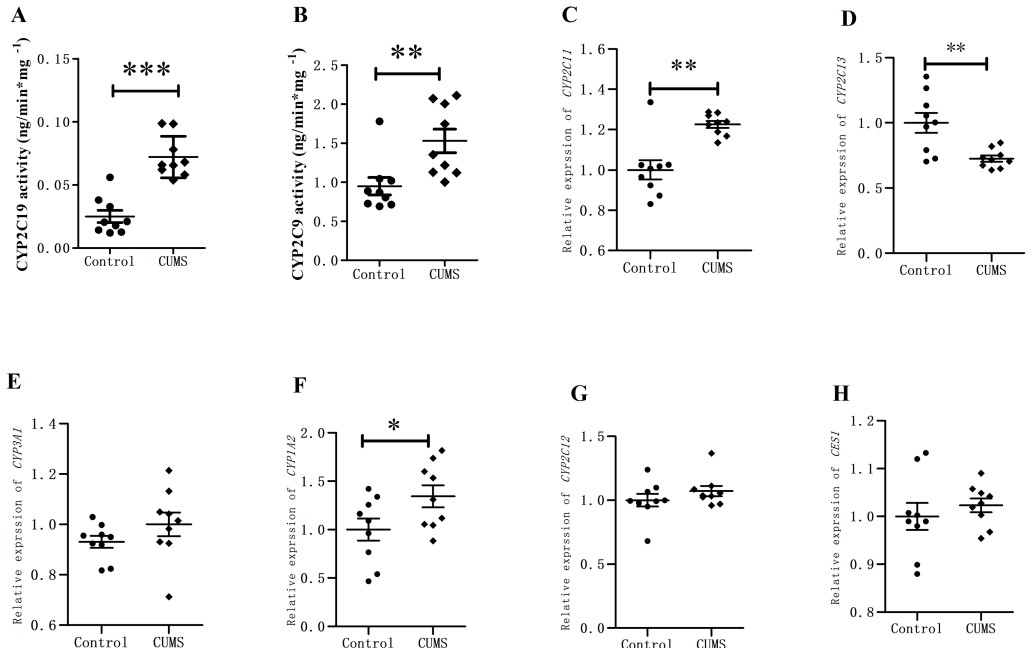

**Figure 5  CYP450 activity and expression in rat livers.** (A) CYP2C19, (B) CYP2C9 activity in in rat liver microsomes, (C) *CYP2C11*, (D) *CYP2C13*, (E) *CYP3A1*, (F) *CYP1A2*, (G) *CYP2C12*, and (H) *CES1* mRNA level (normalized to control), ($n = 9$, mean $\pm$ SD). $^*p < 0.05$, $^{**}p < 0.01$ compared with the control group.

model not only reduced the intensity of stress and retains a variety of stress methods, but also applied the same stress methods in a completely random order in the whole process of stress modeling. The model of *Papp, Willner & Muscat (1991a)* prevented experimental animals from producing predictability and tolerance.

Our modified CUMS model removed the two non-psychosocial environmental stress methods including fasting and water deprivation, which could change the metabolic capacity of the liver. The whole stress did not affect the normal metabolic regulation of rats. The behavioral performance of rats in this modified CUMS depressed model was very similar to the symptoms of lack of interest and decline of social ability of depression patients (*Zeng et al., 2014*). Open field test responded to the horizontal movement ability and vertical exploration ability of rats after stress. Rats were afraid of the new open environment and mainly move in the surrounding area, but less in the central area. However, the exploratory characteristics of rats promoted their motivation to move in the central area. So stress induced depression rats' horizontal movement and inquiry ability decreased, and freezing time in open field test prolonged.

CUMS is a classic animal model for depression-related behaviors in rats (*Qiao et al., 2016*). Behavioural findings show that sucrose intake is decreased in both sexes, but is more affected in male than in females rats exposed to chronic mild stress (*Dalla et al., 2010*). In Willner's model, exposure of animals to 7–13 mild stressors up to 3 months produced a longer lasting depression-like behavior, anhedonia (*Willner et al., 1987*; *Papp, Willner & Muscat, 1991a*; *Willner, 1997*). The present experiment raised female rats in

isolation and increased the sensitivity to depression (*Westenbroek et al., 2004*). The sucrose preference test showed that the sucrose intake and self-reward ability decreased moderately in depressed rats. In the open-field test, the freezing time of depressed rats was prolonged, while the average movement speed and distance were reduced, and the number of times entering the central area was reduced significantly. The motor activities and exploration decreased considerably, indicating the depression-like behavior in depressed rats. These results illustrated that the depression model of SD female rats was successfully established based on the classic chronic unpredictable stress in our study (*Wang et al., 2021*).

Serotonin plays an important role in the behaviours disrupted in depression, such as mood, sleep, and appetite. The serotonergic neurochemical responses are differentially affected in males and females, ultimately producing sex-dependent behavioural effects (*Dalla et al., 2010*). In rats, it has been shown that there is an increase in 5-HT activity, synthesis and metabolites in the females (*Carlsson & Carlsson, 1988*; *Haleem, Kennett & Curzon, 1990*). Conversely, a decrease in brain 5-HT concentration can cause a relapse of depression (*Li et al., 2019*). In our study, the level of cortex 5-HT decreased in depressed females rats, verifying the successful establishment of the CUMS—induced depression rats model.

Clopidogrel is an oral irreversible P2Y12 receptor antagonist which inhibits platelet aggregation and is widely used for patients with acute coronary syndromes or after percutaneous coronary intervention (*Pereira et al., 2020*). Previous research found that a 10 mg/kg clopigrel dose induced gastric injury in rats (*Wu et al., 2020a*). There are many reasons leading to gastric mucosal injury, but the pharmacokinetic process of clopidogrel is a vital factor (*Zhang et al., 2021*). In our study, the plasma concentration of clopidogrel between the depression and control group was determined by HPLC-MS/MS. The peak times for clopidogrel and IS were controlled within 5 min, and the LLOQ was of 0.35 ng mL$^{-1}$, fully meetting the requirements of pharmacokinetic experiment. After single clopidogrel administration, the $AUC_{(0-\infty)}$ and $AUC_{(0-t)}$ of clopidogrel decreased significantly ($*p < 0.05$), while Vz/F and CLz/F increased significantly ($**p < 0.01$). Disturbance of these parameters indicated that CUMS induced depression in SD female rats changed the pharmacokinetics of clopidogrel *in vivo*. The plasma concentration of clopidogrel in the depression group was lower than that in the control group, indicating that clopidogrel metabolism in the depression group was faster than that in the control group. The plasma AUC of rats in the depression group decreased significantly, indicating that the absorption rate decreased. The average residence time *in vivo* also decreased significantly, and the clearance rate accelerated, indicating that the elimination rate of clopidogrel accelerated. In conclusion, the absorption of clopidogrel decreased, metabolism accelerated, distribution increased, and half-life did not change in the depression group. Previous studies found that CUMS-induced depressive SD rats could alter the pharmacokinetics of repaglinide and sitagliptin, although the specific drug metabolic parameters were different (*Xia et al., 2016*; *Wei et al., 2017*).

Clopidogrel is a prodrug, and needs to be converted into active metabolites through CYP450 enzymes in the liver. The main metabolic enzymes involved in the metabolism of clopidogrel in humans are CYP2C19, CYP2C9, CYP1A2 and CYP3A4 (*Chetty, Ravenstijn*

& *Manchandani, 2021*). CYP2C19 is the most important enzyme involved in clopidogrel metabolism. CYP2C19*2 allele is associated with low clopidogrel reactivity, while the CYP2C19*17 allele is not only a protective factor for ischemic events but also a risk factor for bleeding (*Saiz-Rodríguez et al., 2019*). The homologous CYP450 enzyme subtypes of rats and humans are CYP2C19, CYP2C9 and CYP1A2 (*Li et al., 2014*). The amino acid sequence of human CYP2C9 has the most homology to that of rat CYP2C11/13 (*Wang et al., 2009*). In female rats, CYP2C12 accounts for more than 40% of the total content of cytochrome P450 in the liver of female rats (*Banerjee, Das & Shapiro, 2013*). CYP 3A1/2 is the main form of liver drug metabolism enzyme in rats, and it has the same function as CYP3A4 in human beings (*Resham et al., 2015*). CYP1A2 is also involved in the metabolism of clopidogrel, and its genetic polymorphism can lead to drug interaction between clopidogrel and other drugs (*Wang et al., 2015*). Bleeding events after taking clopidogrel are significantly more common among black CYP1A2*1C carriers with acute myocardial infarction (*Cresci et al., 2014*). Eighty-five to ninety percent of clopidogrel is hydrolyzed and metabolized into inactive substances by the carboxylesterase-1 (CES1) (*Chetty, Ravenstijn & Manchandani, 2021*). Therefore, slight disturbance in the expression of the CES1 enzyme can also affect the activity and efficacy of clopidogrel.

In the present study, rat liver microsomes were collected and incubated with substrates. The results showed that the activities of CYP2C19 and CYP2C9 increased significantly in the CUMS group compared with those in the control group, especially that of CYP2C19. A recent study also showed that stress altered CYP2C19 activity (*Zemanova, Anzenbacher & Anzenbacherova, 2022*). Since there were no corresponding primers for CYP2C19 gene expression in the rat liver, the level of CYP2C19 gene expression was not determined. Nevertheless, we found that the *CYP2C11* and *CYP1A2* mRNA expression increased approximately 1.3-fold in the CUMS group, while that of *CYP2C13* mRNA reduced by 27.43%. *CYP3A1* and *CYP2C12* mRNA relative expression increased slightly between two groups, and *CES1* mRNA relative expression did not change in depressed rat livers. Previous studies found that the activity of CYP450 enzymes in the liver microsomes of depressed GK rats induced by CUMS changed significantly (*Xia et al., 2016*). In brief, changes in the expression and activity of CYP450 enzyme disturbed the pharmacokinetic parameters of clopidogrel in CUMS—induced depressed rats.

In clinic, female patients with ACS are prone to comorbid psychological disorders, such as depression (*Koh et al., 2019*). However these patients have a late onset of depression and a high risk of bleeding events after invasive surgery (*Mehilli & Presbitero, 2020*). The females perform in a qualitatively similar manner to males in most tests although there may be sex differences in sensitivity. Tests that utilize conditioned fear paradigms, which involve a learning component appear to be less impacted by the estrous cycle although sex and cycle-related differences in responding can still be detected (*Lovick & Zangrossi, 2021*). In this study, rats were subjected to nine different chronic unpredictable stresses for 3 months, inducing activity and expression changes in CYP450 enzymes in female rats, but there was no significant change in the expression of *CYP2C12* in these rats. The perturbation of clopidogrel pharmacokinetic parameters in depressed rats was significantly influenced by CYP450 enzyme activity and expression. Therefore, the disturbance of CYP450 enzyme

expression and activity in the liver of CUMS rats has a certain reference significance for applying clopidogrel in depressed female patients. However, the current study has some limitations. First, there were certain individual differences in rats in their response to chronic unpredictable stress. Second, administering the antiplatelet drug clopidogrel in rats resulted in death during blood collection.

## CONCLUSIONS

In conclusion, to the best of our knowledge, the present study is the first time to confirm that CUMS-induced depression perturbed the liver CYP450 enzyme expression and activity, disturbing the pharmacokinetics of clopidogrel in female SD rats. This phenomenon can be extrapolated to depressed patients. Therefore, it is recommended to analyze the concentration of clopidogrel in female patients comorbid with depression to prevent the change of drug exposure that result in adverse bleeding events.

### Funding

This work was supported by the Project of Health Commission of Hunan Province (202113012015), the Natural Science Foundation of Changsha Municipality (kq2014203), and the Hunan provincial University Reform and Development Foundation (2050205). The funders had no role in study design, data collection and analysis, decision to publish, or preparation of the manuscript.

### Grant Disclosures

The following grant information was disclosed by the authors:
Project of Health Commission of Hunan Province: 202113012015.
Natural Science Foundation of Changsha Municipality: kq2014203.
Hunan provincial University Reform and Development Foundation: 2050205.

### Competing Interests

The authors declare there are no competing interests.

### Author Contributions

- Xueyao Jiang performed the experiments, prepared figures and/or tables, and approved the final draft.
- Jing Wu performed the experiments, prepared figures and/or tables, and approved the final draft.
- Boyu Tan analyzed the data, prepared figures and/or tables, and approved the final draft.
- Sulan Yan performed the experiments, analyzed the data, authored or reviewed drafts of the article, and approved the final draft.
- Nan Deng conceived and designed the experiments, authored or reviewed drafts of the article, and approved the final draft.
- Hongyan Wei conceived and designed the experiments, prepared figures and/or tables, authored or reviewed drafts of the article, and approved the final draft.

## Animal Ethics

The following information was supplied relating to ethical approvals (i.e., approving body and any reference numbers):

Hunan Provincial People's Hospital / First Affiliated Hospital of Hunan Normal University

Medical ethics committee.

## Data Availability

The raw data is available in the Supplementary File.

## Supplemental Information

Supplemental information for this article can be found online at http://dx.doi.org/10.7717/peerj.14111#supplemental-information.

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
