# Peer review of "Effect of chronic unpredicted mild stress-induced depression on clopidogrel pharmacokinetics in rats"

_PeerJ, doi:10.7717/peerj.14111_

## Round 0.1 · original submission · Major Revisions

The reviewers have recommended publication, but also suggest some major revisions to your manuscript. Therefore, I invite you to respond to the reviewers' comments and revise your manuscript.

Once again, thank you for submitting your manuscript to PeerJ and I look forward to receiving your revision as soon as possible.

·

Basic reporting

This is an interesting study which showed that CUMS-induced depression altered the drug metabolic process of clopidogrel, and the change of CYP450 expression in depressive rats might contribute to the disturbance of clopidogrel pharmacokinetics.

However, the authors did not investigate the underlying mechanisms of why liver cytochrome metabolic enzymes could lead to a significant change in the efficacy of clopidogrel.

With the death of three rats, the sample size of 6 in the CUMS group may be too small to acquire reliable statistical results.

There are many grammar mistakes and typos. Please check through the manuscript. E.g., “modle” (should be “model”), “adlibitum” (“ad libitum”), “mightbe” (“might be”), etc. And there should be one space before brackets.

Experimental design

The authors did not investigate the underlying mechanisms of why liver cytochrome metabolic enzymes could lead to a significant change in the efficacy of clopidogrel.

Validity of the findings

With the death of three rats, the sample size of 6 in the CUMS group may be too small to acquire reliable statistical results.

Reviewer 2 ·

Basic reporting

No comment.

Experimental design

1. The experiment design is not clear. How many groups you have? When and how often you treated the rats with clopidogrel? The order of behavior tests? Please draw a schematic diagram of the experimental design.
It looks like there should be four groups totally but not two.
2. Have you take the effect of estrous cycle of on the results of behavior test into consideration?
3. Patient with ASC undergoing PCI need to take it for 6-12 months. So why you treat rats with clopidogrel only one time?

Validity of the findings

1. Please show all nine individual dots in every bar graph. And most of time we would like to show control data first and then the experimental group.
2. It’s hard to read the axis label of figure 2 & 3. And please keep the font and size of text of each figure consistents.
3. Why you didn’t choose tail suspension test or force swimming test? Open field test is not a typical behavioral test for stress induced depression. Is freezing time in OFT meaningful for stress-induced depression?

Additional comments

1. There is no space between a lot of text and parentheses in the text. The font and format of “P” are not uniform. Please check the whole manuscript carefully.
2. In result part, the data of behavioral test should be represent more clearly, like sucrose preference (Control 81.71… vs. CUMS 68.16…., p<….)

---

## Round 0.2 · Minor Revisions

Dear Authors,

I have now received the reviewers' comments on your manuscript. The reviewers have recommended publication, but also suggest some minor revisions to your manuscript. Therefore, I invite you to respond to the reviewers' comments and revise your manuscript.

Sincerely,

·

Basic reporting

The authors have successfully addressed my comments.

Experimental design

The authors have successfully addressed my comments.

Validity of the findings

The authors have successfully addressed my comments.

Reviewer 2 ·

Basic reporting

no comment

Experimental design

1. The author added a schematic diagram of the experimental design. I wonder why you didn’t have the groups treated with vehicle in your design? Please explain.
2. Please added your explain for estrous cycle and behavior test in the manuscript.

Validity of the findings

1. The authors have modified Figure 6 and 7, but please check Figure 1, and modify it as Figure 6 and 7 (with individual dots). Text size in Figure 3 and 4 is still too small to read.
2. The sample size is confusing. As you mentioned, with the death of three rats, it should be six. Why there are nine dots in figure 6 and 7? And in behavior test, it’s also nine in raw data.

Additional comments

1. Information about Table 3 “The distribution volume (Vz/F) and clearance (CLz/F)” showed in legend of Figure 5. It’s confusing.
2. Please reorganize the figures and tables, especially figure 3, 4 and 5. In my opinion, four or five figures totally is enough for this manuscript.
3. English language could be improved more.

---

## Round 0.3 · accepted · Accept

Many thanks for addressing all the issues.